# Neuronal representation of environmental boundaries in egocentric coordinates

James R. Hinman [1,2], G. William Chapman[1] & Michael E. Hasselmo[1]

Movement through space is a fundamental behavior for all animals. Cognitive maps of environments are encoded in the hippocampal formation in an allocentric reference frame, but motor movements that comprise physical navigation are represented within an egocentric reference frame. Allocentric navigational plans must be converted to an egocentric reference frame prior to implementation as overt behavior. Here we describe an egocentric spatial representation of environmental boundaries in the dorsomedial striatum.

---

[1] Center for Systems Neuroscience, Department of Psychological and Brain Sciences, Boston University, 610 Commonwealth Avenue, Boston, MA 02215, USA. [2] Present address: Department of Psychology, University of Illinois Urbana-Champaign, 603 East Daniel Street, Champaign, IL 61820, USA. Correspondence and requests for materials should be addressed to J.R.H. (email: hinmanlab.uiuc@gmail.com) or to M.E.H. (email: hasselmo@bu.edu)

The hippocampus, entorhinal cortex, and associated structures store spatial representations in an allocentric or world-centered reference frame that is strongly influenced by environmental boundaries[1–5]. Computational models suggest that allocentric navigational representations such as boundary responses[4–6] must arise from and be converted back to an egocentric reference frame to guide overt behavior[7–9]. The dorsomedial striatum (DMS) shows neural responses related to action decisions[10–12], plays a critical role in controlling behavioral output[11] including egocentric navigational strategies[13], and receives input from regions involved in spatial navigation including medial entorhinal and retrosplenial cortex[14].

## Results

**Egocentric boundary coding in DMS**. To determine whether egocentric spatial information is present in the DMS, male Long–Evans rats ($n = 4$) were implanted with up to 16 tetrodes targeting DMS (Fig. 1a, Supplementary Fig. 1) and single units were recorded ($n = 939$ single units in $n = 44$ sessions) while rats foraged for randomly scattered food in a familiar open field arena.

Stable head direction cells (HDCs, $n = 31$) were found, similar to previous results in the striatum[15] and other structures[16], but few cells had allocentric spatial correlates ($n = 19$ spatially stable cells; Supplementary Fig. 2). However, cells were observed with activity restricted to the environment perimeter only when the rat moved with a particular orientation relative to the walls, suggesting an egocentric coding scheme for boundaries.

To assess the possibility of such an egocentric representation, we created egocentric boundary ratemaps (Fig. 1b, Supplementary Fig. 3) that illustrate the orientation and distance of the boundaries relative to the rat's movement direction (rather than head direction; Fig. 1g, Supplementary Fig. 3) when a cell spikes. Eighteen percent of recorded cells (171/939 cells) were identified with significant firing when a boundary occupied a specific orientation and distance relative to the animal based on the mean resultant length (MRL) of boundary directional firing exceeding the 99th percentile of a shuffled distribution (Fig. 1f) and responding stably across the two halves of a recording (Supplementary Fig. 3l, m). We termed these egocentric boundary cells (EBCs; EBCs per animal: mean = 42.75, range = 15–70; Fig. 1c, d, Supplementary Fig. 4). A subset of EBCs had firing

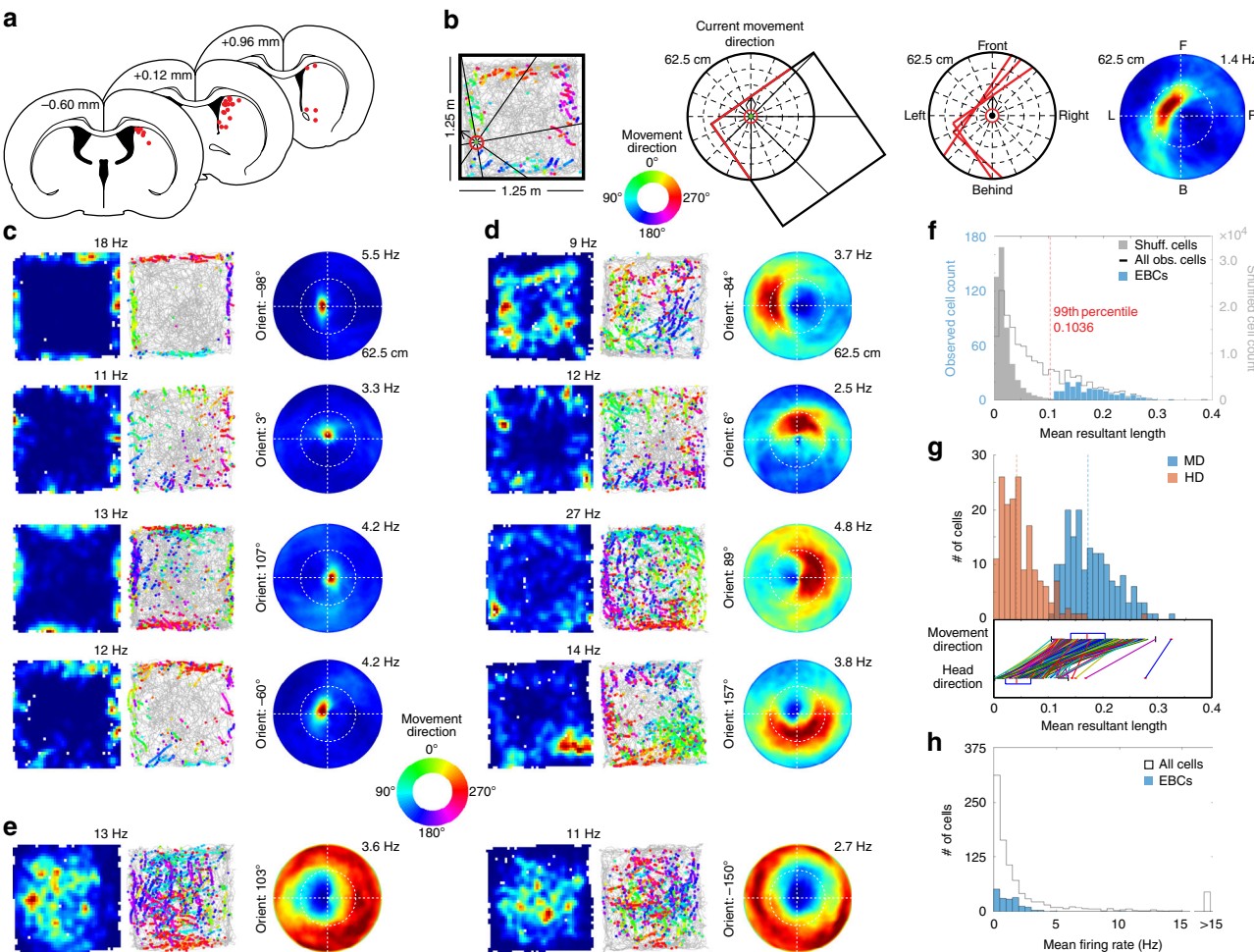

**Fig. 1** Dorsomedial striatal cells respond to environmental boundaries in an egocentric reference frame. **a** Final tetrode positions. Diagrams were reprinted from The rat brain in stereotaxic coordinates, 7th edition, Paxinos, G. & Watson, C., pages 97, 111 and 123, Copyright 2014, with permission from Elsevier[26]. **b** Egocentric boundary ratemap generation (see Methods and Supplementary Fig. 3). **c** Columns show allocentric spatial ratemaps, trajectory plots with color-coded movement direction spike locations (color wheel legend shown in figure), and egocentric boundary ratemaps for four example egocentric boundary cells (EBCs) with different preferred orientations and a preferred distance close to the animal. The maximum firing rate for each allocentric and egocentric ratemap is displayed above the top right corner of each plot. **d** Same as in **c**, but for EBCs with preferred distances distant from the animal. **e** Same plots as above, but for two inverse EBCs. **f** Distribution of mean resultant length for observed and shuffled cells. **g** Distribution of mean resultant lengths for EBCs using movement direction and head direction. **h** Distribution of mean firing rates

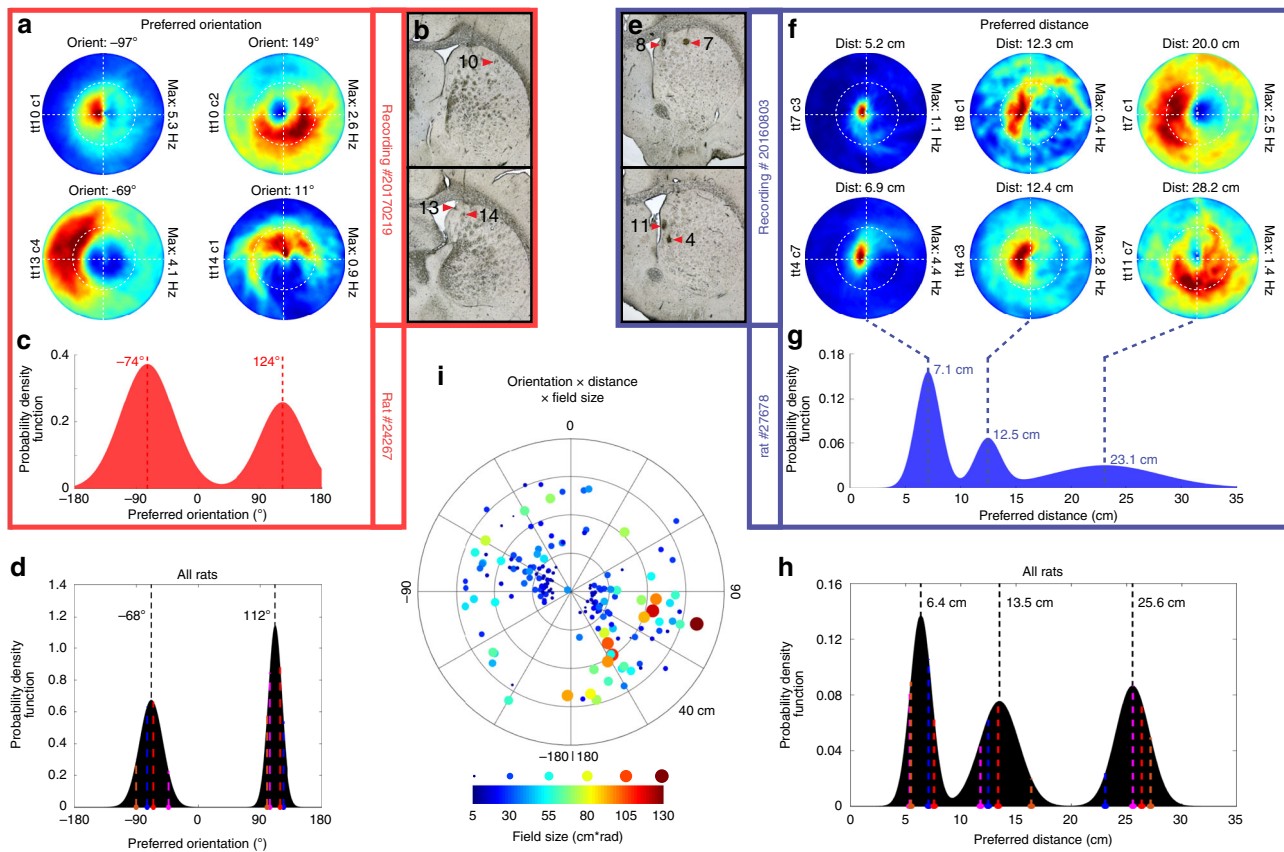

**Fig. 2** Preferred orientation and distance of egocentric boundary cells (EBCs). **a** Egocentric boundary ratemaps for four simultaneously recorded EBCs with different preferred orientations, specified above each plot, tetrode (tt) number and cell (c) number displayed to the left of each plot and the maximum firing rate displayed to the right of each plot. **b** Final tetrode locations for cells shown in **a**. Numbers indicate the tetrode number. **c** Probability distribution function of preferred orientation for all EBCs from a single rat. **d** Probability distribution function of EBC preferred orientation peaks from all rats. Colored dashed lines indicate peak locations from each animal. **e** Final tetrode locations for cells shown in **f**. **f** Egocentric boundary ratemaps for six simultaneously recorded EBCs with different preferred distances specified above each plot, tetrode (tt) number and cell (c) number displayed to the left of each plot and the maximum firing rate displayed to the right of each plot. **g** Probability distribution function of preferred distance for all EBCs from a single rat. **h** Probability distribution function of EBC preferred distance peaks from all rats. Colored dashed lines indicate peak locations from each animal. **i** Polar scatter plot of preferred distance vs. preferred orientation for all EBCs with field size represented with both color and dot diameter

rates that decreased in response to a boundary (n = 49; Fig. 1e) that we termed inverse EBCs (iEBCs). EBCs and iEBCs had low mean firing rates (mean ± SEM: 1.26 ± 0.09 Hz, n = 171 cells; Fig. 1h) and virtually all (97%) fired phasically consistent with them being DMS medium spiny neurons[17].

The population of EBCs responds to boundaries at the full spectrum of orientations relative to the animal, although the distribution of preferred orientation is bimodal with peaks sitting 180° opposite each other on either side of the animal (−68° and 112°; Fig. 2, Supplementary Fig. 5), while being slightly offset from perpendicular to the animal's long axis by 22° (Fig. 2d). The offset did not result from a bias in the boundary approach trajectories of the animals (Supplementary Fig. 5), but may stem from a lateralized cortical representation upstream of EBCs and the largely ipsilateral nature of cortico-striatal projections. The distribution of preferred boundary distance contained three peaks (6.4, 13.5, and 25.6 cm) indicating the presence of three distinct preferred distances among EBCs (Fig. 2f–h, Supplementary Fig. 5) that could be important for a hierarchical navigation search strategy on multiple scales similar to grid cells. The size of EBC receptive fields increased as a function of preferred distance (Pearson's correlation: r = 0.46, p < 2.07e−10; Fig. 2i), indicating that the egocentric boundary representation has greater precision the closer the animal is to the boundaries. Both preferred

orientation and distance lacked clear topography given that EBCs with different orientations and distances appeared on the same tetrode (Fig. 2a, b, e, f).

**EBCs respond stably to local boundaries across environments.** To confirm that EBCs respond to local boundaries rather than distal features of the testing room, we conducted recordings after rotating the open field with four black walls 45° relative to the testing room with numerous static extra-maze cues, putting local boundaries and testing room boundaries maximally out of alignment. Recordings were obtained in the standard and rotated open field orientation (n = 130 cells; n = 4 sessions), including 19 EBCs and 3 HD cells. Following the rotation both the preferred orientation (Wilcoxon's signed rank test: z-score: −0.63, n.s., z-score: 1.99, n.s.; Fig. 3a, e, Supplementary Fig. 6) and preferred distance (Wilcoxon's signed rank test: z-score: 0.19, n.s., z-score: −1.16, n.s.; Fig. 3a, e, Supplementary Fig. 6) of EBCs remained unchanged. In contrast, HD cells remained anchored to the overall testing room (Supplementary Fig. 6). Given the salience of corners, we considered the possibility that EBCs uniquely code these local environmental attributes and identified a subset of EBCs (n = 16; 9.4%) with firing rate differences near the corners compared to the middle of the boundaries. This indicates that

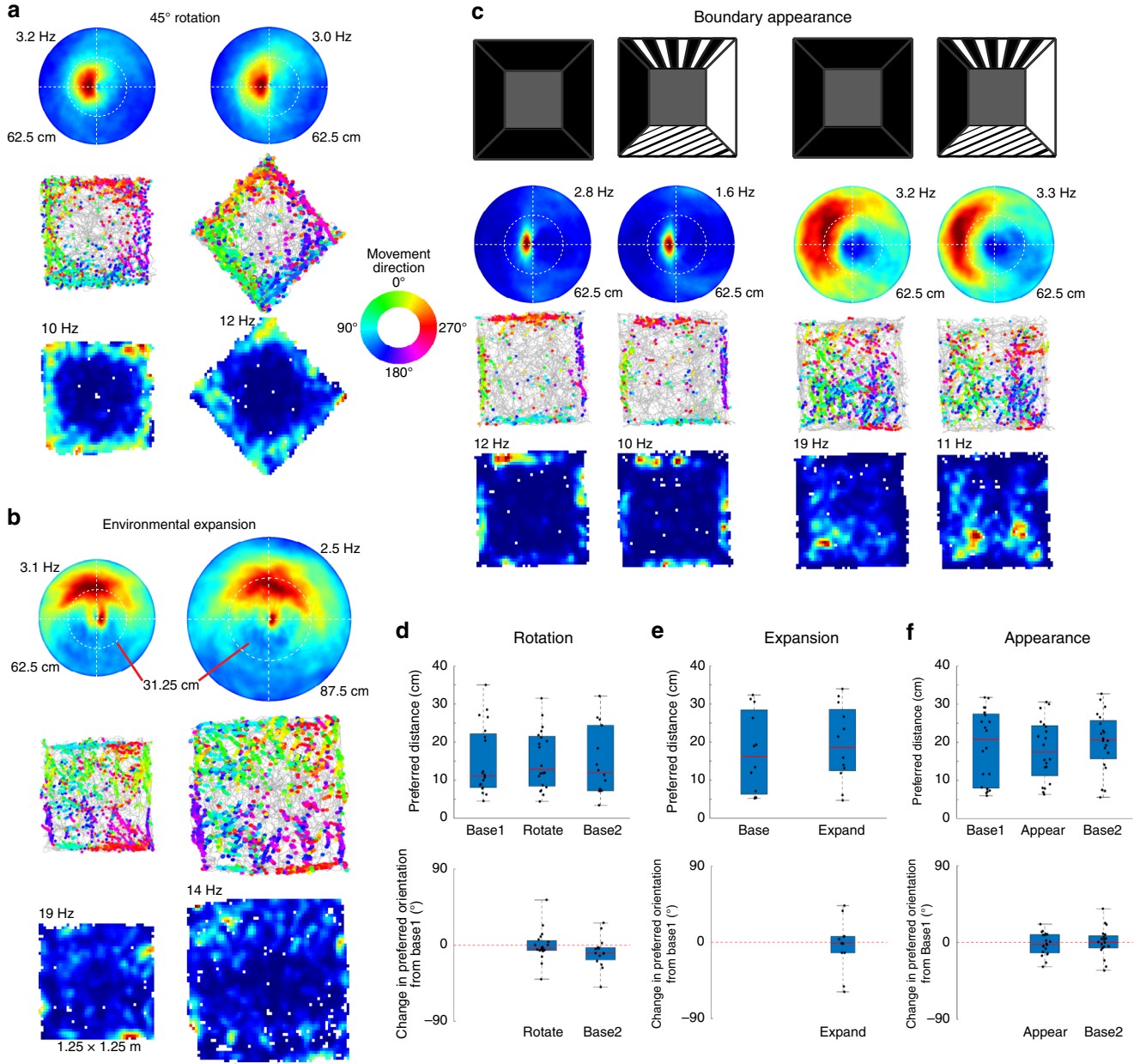

**Fig. 3** Egocentric boundary cells (EBCs) respond stably to local environmental boundaries. **a** Egocentric boundary ratemaps with maximum firing rates indicated above each plot, trajectory plots with color-coded movement direction spike locations, and allocentric spatial ratemaps with maximum firing rates indicated above each plot (top to bottom) for an EBC during a baseline and 45° environmental rotation session (left to right). **b** Same plots as in **a**, but for the standard 1.25 × 1.25 m² environment and an expanded 1.75 × 1.75 m² environment. **c** Same plots as in **a**, but for two EBCs (left to right) during recordings with the standard black walls and patterned walls. **d–f** Box plots of preferred distance (top) and change in preferred orientation from baseline (bottom) where the top and bottom of each box represent the first and third quartile, the red line indicates the median and the whiskers indicate the full range of values for (**d**) the rotation experiment (preferred distance: Base1: 25th: 8.05, 50th: 11.06, 75th: 22.16 cm, Rotate: 25th: 8.42, 50th: 12.88, 75th: 21.54 cm, Base2: 25th: 7.24, 50th: 12.02, 75th: 24.42 cm; preferred orientation: Rotate: 25th percentile: −6.15, 50th percentile: −3.21, 75th percentile: 5.28° change from Base1, Base2: 25th: −17.34, 50th: −9.30, 75th: −2.80° change from Base1), (**e**) the expansion experiment (preferred distance: Base: 25th: 6.33, 50th: 16.20, 75th: 28.46 cm; Expand: 25th: 12.51, 50th: 18.68, 75th: 28.58 cm; preferred orientation: Expand: 25th: −12.30, 50th: −1.02, 75th: 7.02° change from Base), (**f**) the visual appearance experiment (preferred distance: Base1: 25th: 7.96, 50th: 20.74, 75th: 27.44 cm; Pattern1: 25th: 11.21, 50th: 17.42, 75th: 24.37 cm; Base2: 25th: 15.65, 50th: 20.67, 75th: 25.69 cm; preferred orientation: Pattern1: 25th: −12.10, 50th: −2.04, 75th: 9.40° change from Base1; Base2: 25th: −6.30, 50th: 0.40, 75th: 7.80° change from Base1)

EBCs respond to local boundaries of the open field rather than the larger recording room and that egocentric and allocentric reference frames can be computed in parallel[7,8].

To further test EBC responses across different environmental manipulations, we performed recordings in open fields of different sizes, which provides information regarding whether

EBCs have a constant preferred distance or instead scale with environment size. Recordings were obtained ($n = 50$ cells, $n = 12$ EBCs; $n = 3$ sessions) in open fields with walls differing in length by 50 cm. Regardless of the size of the open field, EBCs responded to boundaries at the same distance (Wilcoxon's signed rank test: z-score: −0.71, n.s.; Fig. 3b, f, Supplementary Fig. 7) and

orientation (Wilcoxon's signed rank test: $z$-score: $-0.31$, n.s.; Fig. 3b, f, Supplementary Fig. 7) from the animal, indicating a lack of scaling with environment size.

The striatum receives input from several visual cortical regions[14]. Therefore, we asked whether boundary appearance influenced EBC responses. Recordings were performed ($n = 73$ cells, $n = 19$ EBCs; $n = 4$ sessions) in an environment with four black walls and then with three of the walls swapped with walls of different patterns (Fig. 3c, Supplementary Fig. 8). The firing fields of EBCs did not change in either preferred orientation (Wilcoxon's signed rank test: $z$-score: $-0.63$, n.s., $z$-score: $0.19$, n.s.; Fig. 3c, g, Supplementary Fig. 8) or preferred distance (Wilcoxon's signed rank test: $z$-score: $-0.15$, n.s., $z$-score: $-0.63$, n.s.; Fig. 3c, g, Supplementary Fig. 8) with the change in visual appearance of the walls. The lack of effect of wall visual appearance on EBCs suggests a higher-order representation of a boundary independent of basic visual features.

The allocentric cognitive map of a given environment maintained in the hippocampal formation is stable over time. We tested whether EBCs maintain a stable representation of a given environment over time by performing two recordings ($n = 426$ cells, $n = 80$ EBCs; $n = 19$ sessions) in the same open field. Both the preferred orientation (Wilcoxon's signed rank test: $z$-score: $-0.87$, n.s.; Supplementary Fig. 9) and preferred distance (Wilcoxon's signed rank test: $z$-score: $-0.75$, n.s.; Supplementary Fig. 9) of EBCs remained stable across sessions. Given the stability of EBC representation for a single environment, we next tested whether EBCs remap across environments as does the allocentric spatial map[18,19]. Recordings were obtained ($n = 38$ cells, $n = 14$ EBCs; $n = 2$ sessions) as rats explored a familiar and completely novel open field in a novel testing room. EBCs responded with the same preferred orientation (Wilcoxon's signed rank test: $z$-score: $1.35$, n.s., $z$-score: $-2.04$, n.s., $z$-score: $-0.66$, n.s.; Fig. 4a, c, Supplementary Fig. 10) and distance

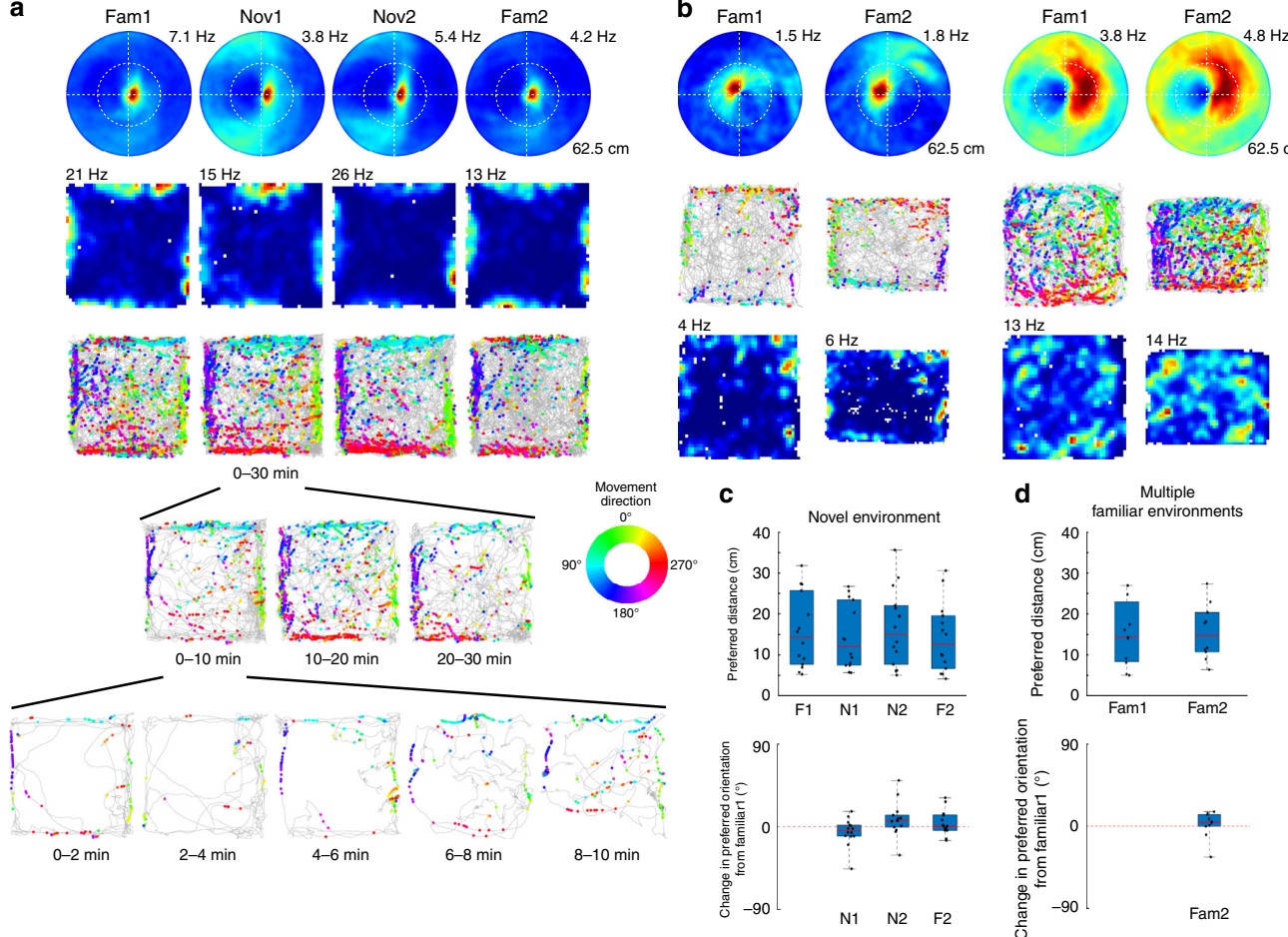

**Fig. 4** Egocentric boundary cells (EBCs) do not remap across environments. **a** Egocentric boundary ratemaps with maximum firing rates indicated above each plot, allocentric spatial ratemaps with maximum firing rates indicated above each plot, and trajectory plots with color-coded movement direction spike locations (top to bottom) for an EBC recorded in a familiar and novel environment (left to right). **b** Egocentric boundary ratemaps with maximum firing rates indicated above each plot, trajectory plots with color-coded movement direction spike locations, and allocentric spatial ratemaps with maximum firing rates indicated above each plot (top to bottom) for two example EBCs recorded in different familiar environments. **c** Box plots of preferred distance (top) and change in preferred orientation from baseline (bottom) where the top and bottom of each box represent the first and third quartile, the red line indicates the median and the whiskers indicate the full range of values for the novel environment experiment (preferred distance: Fam1: 25th: 7.68, 50th: 14.32, 75th: 25.73 cm; Nov1: 25th: 7.49, 50th: 12.06, 75th: 23.45 cm; Nov2: 25th: 7.68, 50th: 15.04, 75th: 22.07 cm; Fam2: 25th: 6.67, 50th: 12.62, 75th: 9.58 cm; preferred orientation: Nov1: 25th: $-10.22$, 50th: $-4.09$, 75th: $1.86°$ change from Fam1; Nov2: 25th: $-0.30$, 50th: 6.90, 75th: $12.74°$ change from Fam1; Fam2: 25th: $-3.91$, 50th: 0.92, 75th: $12.88°$ change from Fam1). **d** Same plots as in **c** but for the multiple familiar environment experiment (preferred distance: Fam1: 25th: 8.38, 50th: 14.33, 75th: 22.98 cm; Fam2: 25th: 10.75, 50th: 14.80, 75th: 20.39 cm; preferred orientation: Fam2: 25th: $-0.23$, 50th: 4.43, 75th: $12.85°$ change from Fam1)

(Wilcoxon's signed rank test: $z$-score: 1.41, n.s., $z$-score: $-0.97$, n.s., $z$-score: 0.28, n.s.; Fig. 4a, c, Supplementary Fig. 10) immediately upon exposure to the novel environment (Fig. 4a, bottom) and continued to do so for at least the first two exposures to a novel environment. It is possible that EBC representations of different environments could diverge with additional experience, and therefore, recordings were obtained from two familiar, but different open field environments ($n = 42$ cells, $n = 11$ EBCs; $n = 3$ sessions). The additional experience did not result in the divergence of the EBC representations, as neither the preferred orientation (Wilcoxon's signed rank test: $z$-score: $-1.07$, n.s.; Fig. 4b, d) nor distance (Wilcoxon's signed rank test: $z$-score: 0.36, n.s.; Fig. 4b, d) changed between the two familiar open fields. Thus, the striatal EBC representation does not remap across environments, but instead provides a stable representation of environmental boundaries relative to the animal across environments.

## Discussion

The present work identified an egocentric representation of environmental boundaries consistent with theoretical predictions from computational models[7,8], which propose that allocentric spatial representations in the hippocampal formation are generated from and converted back to an egocentric spatial representation prior to behavioral output. The striatum receives input from navigation related cortical structures[14], including medial entorhinal, retrosplenial, and posterior parietal cortex, but does not directly project to these structures. This suggests that the striatal egocentric representation is not directly involved in the generation of the allocentric spatial representation, while it remains unknown whether the allocentric spatial map is needed to generate the striatal egocentric representation.

Allocentric spatial information could be transformed into an egocentric representation through a process involving allocentric boundary coding cells in medial entorhinal cortex[5] and subicular cortices[4,6], postsubicular cells with mixed allocentric-egocentric coding[20], and egocentric cells in lateral entorhinal cortex[9]. Additionally, EBCs in DMS are more strongly coupled to the movement direction of the animal than the animal's head direction, and movement direction has been proposed to be a potential output of the grid cell network in MEC[21]. Alternatively, cortical regions such as posterior parietal cortex that contain egocentric sensory and motor representations[22–24] could be involved in generating the striatal egocentric representation without allocentric input. The retrosplenial cortex has been proposed as a potential locus for reference frame transformations[7,8] given its efferent and afferent connections with structures that utilize allocentric and egocentric reference frames, while itself utilizing conjunctive allocentric, egocentric, and route-based coding[25], and therefore may be an important source of information for generating EBCs in DMS. The data presented here indicates an important egocentric neural representation of boundaries that could interact with allocentric coding of environmental boundaries to guide movement in egocentric coordinates.

## Methods

Subjects: Male Long–Evans rats ($n = 4$) obtained from Charles River Labs (Wilmington, MA, USA) were individually housed in plexiglass cages in a temperature- and humidity-controlled facility with a 12 h/12 h light/dark cycle. Animals had free access to food and water prior to the initiation of all experiments. All procedures were approved by the Institutional Animal Care and Use Committee at Boston University.

Presurgical procedures: At the start of all experiments, animals were acclimated for at least 2 days to the experimental testing room, to being handled by the researcher, and to eating crushed Froot Loops (General Mills, Battle Creek, MI, USA), which served as the food that animals searched for in the open field. For a minimum of 3 days following that acclimation, animals were exposed to the familiar open fields for up to 20 min/day. The primary familiar open field was $1.25 \times 1.25$ m$^2$ with black walls 36 cm in height and a dark gray textured rubber floor. The secondary familiar open field was $1.25 \times 1.00$ m$^2$ with grays walls 72 cm in height relative to a black particle board floor that was raised 54 cm off of the floor.

Surgical procedures: Aseptic surgery was performed for the implantation of a custom-built 12 tetrode hyperdrive targeting the medial striatum. Surgery began with the administration of atropine (0.1 mg/kg) subcutaneously and then anesthesia was induced with a combination of a ketamine cocktail (ketamine: 12.92 mg/kg; acepromazine: 0.1 mg/kg; xylazine: 1.31 mg/kg) administered intraperitoneally and isoflurane administered via inhalation at an initial concentration of 0.5%. Upon loss of a toe pinch reflex the rat's head was shaved and the animal was positioned in a stereotaxic frame. A midline incision was made in the rat's scalp and any connective tissue covering the skull was cleared. Anchor screws were positioned across the skull surrounding the implantation site and the ground screw was positioned over the cererbellum. A craniotomy centered at the hyperdrive implantation site over the medial striatum (A/P: + 0.5; M/L: 2.5) was made allowing the medial edge of the hyperdrive to be positioned close to the medial border of the striatum given an approximate 2 mm inner diameter of the cannula housing the tetrodes. Upon completion of the craniotomy, dura was resected and the hyperdrive was lowered until the cannula housing the tetrodes contacted the dorsal surface of the brain. The remaining space in the craniotomy was filled with Kwik-Sil (World Precision Instruments, Sarasota, FL, USA) and the hyperdrive was secured in place by connecting it to the anchor screws with dental cement. Once the hyperdrive was secured in place, the tetrodes were each lowered into the brain. The tetrodes in the first two animals were lowered approximately 4.25 mm, while in the subsequent three animals tetrodes were lowered approximately 2 mm to the deep layers of cortex. Post-operative antibiotics (Baytril: 10.0 mg/kg) and analgesics (Ketafen: 5.0 mg/kg) were administered for 5 days and animals were allowed to recover for 7 days with free access to food and water prior to any involvement in experiments.

Electrophysiological recordings: At the start of each day, the rat was placed on an elevated pedestal where it was connected to the electrophysiological acquisition equipment. Neural signals were initially amplified via two headstages attached to a single 64-channel electrode interface board prior to being transmitted to the 64-channel Digital Lynx SX acquisition system (Neuralynx, Bozeman, MT, USA) where the signals were digitized, filtered (0.3–6.0 kHz), and further amplified (5000–20,000×). Spikes were detected online as a threshold crossing on any of the channels of a tetrode, at which point a window around the threshold crossing time point from each channel of the tetrode was stored for later analysis. Following each experiment, spikes were assigned to individual single units offline using Offline Sorter (Plexon Inc., Dallas, TX, USA). The peak, peak-to-valley, and principal components of the waveforms were utilized for sorting the spikes. The position of the animal was tracked during the recording through the use of a camera positioned over the recording arena. A red and a green diode attached to the headstage were tracked in order to obtain both the animal's position and head direction throughout the recording.

On any given day, an initial 20 min recording was obtained while the rat foraged for small pieces of Froot Loops (General Mills, Battle Creek, MI, USA) scattered on the floor of a familiar open field. The open fields were always open to the testing room, which had a variety of cues. One of several possibilities followed the initial recording depending upon the assessment of the researcher as to whether any and/or how EBCs were present during the first recording of the day. In some cases, no more recordings were made that day and tetrodes were generally moved ventrally by approximately 70 μm. On some days, in order to assess the stability of EBCs within a single environment a second 20 min recording in the same familiar open field was conducted. On the remainder of days, one of several different manipulations were performed including:

1. Open field rotation: The standard $1.25 \times 1.25$ m$^2$ open field was rotated 45° relative to the testing room. Sessions in the rotated open field were preceded and succeeded by standard open field sessions, except for one experiment where a single standard session and a single rotated session were collected.
2. Open field expansion: The standard $1.25 \times 1.25$ m$^2$ open field could be expanded or contracted to have wall lengths from 1.0 to 1.75 m. Animals were run in square environments with walls that varied by 50 cm (either 1.0 vs. 1.5 m or 1.25 vs. 1.75 m). Recordings performed in the $1.75 \times 1.75$ m$^2$ open field lasted longer in order to obtain adequate spatial coverage of the environment. Expansion experiments minimally included recordings in square environments with walls differing by 50 cm, but also included a recording in the standard $1.25 \times 1.25$ m$^2$ when the 1.0 and 1.5 m long walls were used.
3. Visual appearance: The standard $1.25 \times 1.25$ m$^2$ open field had four black walls that could be swapped for walls with different patterns. Three of the four walls were changed from the standard black walls to one of three different walls including an all white wall, a black wall with thin diagonal white stripes and a wall with black and white vertical stripes of equal widths. A sequence of four recordings was obtained with two sessions in the environment with the patterned walls preceded and succeeded by a session with all black walls. One experiment only included a single session with the patterned walls flanked by sessions with all black walls.

4.  Multiple familiar environments: Animals were run in the standard $1.25 \times 1.25$ m$^2$ open field and then the alternative $1.00 \times 1.25$ m$^2$ familiar open field described above in presurgical procedures. All animals had a minimum of 8 days of exposure to each open field prior to any recordings. One recording in each familiar environment was collected.

5.  Novel open field: Animals were first run in the standard $1.25 \times 1.25$ m$^2$ open field and then brought into an adjacent testing room they had never been in previously. In this novel testing room, rats were recorded while they foraged in a novel square open field that had a smooth white floor and 1.25 m long black walls that were 30.5 cm tall. A sequence of four recordings was collected including two sessions in the novel environment and two sessions in the familiar environment with one preceding and one succeeding the novel environment sessions.

Histology: Animals were deeply anesthetized with isoflurane and small lesions were made at the end of tetrodes that had preliminarily been identified as having EBCs. The lesions were made by passing a small 20 μA current through each channel of the tetrode for 10 s. After the lesions were made, the animals were transcardially perfused with 0.9% saline followed by 10% formalin solution. The brain was then removed from the skull and post-fixed in 10% formalin until it was sliced. The rostral portion of the brain was mounted in a vibratome (Leica Biosystems, Buffalo Grove, IL, USA) where 50 μm coronal sections were taken through the rostral–cuadal extent of the striatum. Slices were mounted on gelatin-coated slides and allowed to dry. Sections from three of the four animals were imaged at this point (Supplementary Fig. 1). Sections were then Nissl stained and cover slipped. The lesions in one animal were noticeably smaller or absent, although tetrode tracks were clearly visible. The locations of lesions or the end of tetrode tracks were identified through the use of light microscopy and images (Supplementary Fig. 1) were obtained using a Nikon DXM1200 camera mounted on an Olympus BX51 light microscope.

Allocentric spatial ratemap generation: The open field arena was divided into equally sized spatial bins ($3 \times 3$ cm$^2$) and the firing rate within each spatial bin was calculated as the number of spikes occurring in a given bin divided by the amount of time the animal spent in the bin. The resulting occupancy normalized two-dimensional (2D) firing rate histograms were smoothed with a 2D 3 cm Gaussian Kernel to generate the final ratemap. The 2D firing rate histogram was color coded from 0 (blue) to the maximum of the firing rate distribution (red) with that maximum firing rate value specified above the top-left corner of each allocentric spatial ratemap. Spatial bins with insufficient occupancy to calculate a firing rate appear white in the ratemaps.

Allocentric trajectory plot generation: The position of the rat obtained by tracking the LEDs attached to the headstage is plotted as a gray line. Each dot indicates the position that the animal occupied when a given cell spiked and is color-coded using the animal's movement direction at the time of the spike. The movement direction is calculated as the instantaneous derivative of the continuous position signal and each figure contains a legend in the form of a colored ring depicting the colors associated with each movement direction.

Egocentric boundary ratemap generation: Egocentric boundary ratemaps were designed based on the same principle as the allocentric spatial ratemap, but instead of considering the data in an allocentric reference frame where the rat's position is considered relative to a static spatial environment (Supplementary Fig. 3a–d), the data were considered in an egocentric reference frame where the boundary position was considered relative to a static rat position (Supplementary Fig. 3e–j). The primary components used to generate the egocentric boundary ratemaps are the animal's movement direction and the boundary position relative to the animal. The instantaneous derivative of the continuous position signal ($x/y$ coordinates) served as the movement direction of the animal. The position of the boundaries relative to the animal was calculated on a frame-by-frame basis. The 360° around the animal was divided into 3° angular bins centered (0°) on the animal's current instantaneous heading and the distance from the animal's current position was divided into 2.5 cm distance bins up to a maximum distance of ½ the length of the longest boundary, yielding 3° × 2.5 cm bins (Supplementary Fig. 3e–j). For each frame, the presence of a boundary in each bin is counted resulting in an egocentric boundary occupancy map (Supplementary Fig. 3i). Then, for a given cell a density plot of boundary location at the time of spiking is generated using the same 3° × 2.5 cm bins (Supplementary Fig. 3h). Occupancy normalized egocentric boundary ratemaps were then generated as the element wise division of the spike density by occupancy, which was then smoothed by a 2D gaussian kernel, with a width of 5 bins and standard deviation of 5 bins (Supplementary Fig. 3j).

A comparison of generating egocentric boundary ratemaps using head direction instead of movement direction affirmed the use of movement direction as the MRLs of boundary direction firing were greater when using movement direction rather than when using head direction for all recorded cells. Restricting consideration to only those cells identified as EBCs when using movement direction, the MRLs were significantly greater when using movement direction than when using head direction (Fig. 1g). Finally, if egocentric boundary ratemaps generated using head direction are used for classifying EBCs, only 17 EBCs are identified, with 7 of those also being classified as EBCs when using movement direction to generate the egocentric boundary ratemaps (Supplementary Fig. 3k). As reported in the main text, when using movement direction to generate egocentric boundary ratemaps, a total of 171 EBCs are identified.

Ratemap dispersion, coherence and field size: Ratemap dispersion was calculated as the mean distance between the ratemap bins with the top 10% of firing rates, while ratemap coherence was calculated as the correlation between the firing rate in each bin and the mean firing rate of all adjacent bins. Receptive field size was calculated as the area bounded by contours surrounding bins with firing rates >75% of the bin with the maximum firing rate.

Egocentric boundary cell classification: The first step in identifying EBCs involved calculating the MRL of each cell's egocentric boundary directional firing independent of boundary distance (Supplementary Fig. 5b). First, the mean resultant was calculated as

$$\text{MR} = \left( \sum_{\theta=1}^{n} \sum_{D=1}^{m} F_{\theta,D} * e^{i*\theta} \right) / (n * m), \qquad (1)$$

where $\theta$ is the orientation relative to the rat, $D$ is the distance from the rat, $F_{\theta,D}$ is the firing rate in a given orientation-by-distance bin, $n$ is the number of orientation bins, $m$ is the number of distance bins, e is the Euler constant, and i is the imaginary constant. Then MRL, used as a measure of boundary orientation specificity, is calculated as

$$\text{MRL} = \text{abs}(\text{MR}) \qquad (2)$$

and the mean resultant angle (MRA), which is used as the preferred egocentric boundary orientation, is calculated as

$$\text{MRA} = \arctan2 \left( \frac{\text{imag}(\text{MR})}{\text{real}(\text{MR})} \right). \qquad (3)$$

The preferred egocentric boundary distance was calculated along each cell's preferred egocentric boundary direction MRA by fitting the firing rate vector along that angle with a Weibull distribution and taking the distance bin with the maximum estimated firing rate as the cell's preferred boundary distance (Supplementary Fig. 5c).

A shuffling procedure was used in order to obtain an MRL significance threshold for identifying EBCs. The spike train of each cell was shifted by random intervals ranging from 30 s to the full length of the recording minus 30 s relative to the behavioral data. Spike times shifted past the end of the behavioral session data were wrapped around to the beginning of the session. The shifted spike trains maintain the same firing statistics as the original spike train, but the spike times have been dissociated from the animal's behavior. For each shifted spike train MRL was calculated and this procedure was bootstrapped 100 times for each cell, thus generating a distribution of MRL values from which the 99th percentile was identified and used as a threshold for identifying significant EBCs (Fig. 1f). In order to ensure that cells maintained a consistent representation throughout the initial recording session, the session was divided into halves and cells were classified as EBCs if all of the following criteria were met: (1) mean firing rate was >0.1 Hz, (2) MRL for both the 1st and 2nd half was greater than the 99th percentile of the shuffled distribution, (3) the change in MRA between the 1st and 2nd half was <45°, and (4) the change in preferred boundary distance between the 1st and 2nd half was <75% of the preferred distance for the whole session. The cells meeting these criteria were then clustered using the $k$-means clustering algorithm using firing rate, boundary direction MRL, MRA, preferred boundary distance, Prop$_{\text{ISIs}>2\,\text{s}}$ (see below), egocentric ratemap coherence, egocentric ratemap dispersion, and field size values ranging from 25 to 85% of the maximum firing rate (10% steps) as features.

HDC classification: The MRL $R_m$ was calculated for each cell as

$$R_m = \frac{\cos(\bar{\theta}) \sum_{i=1}^{n} F_i \cos(\theta_i) + \sin(\bar{\theta}) \sum_{i=1}^{n} F_i \sin(\theta_i)}{\sum_{i=1}^{n} F_i}, \qquad (4)$$

where $\bar{\theta}$ was the head direction of firing and $F_i$ and $\theta_i$ were the firing rate and head direction for bin i. HDCs were identified as those cells with $R_m > 0.3$.

Spatially stable allocentric cell classification: Cells with stable allocentric firing were identified as those cells with allocentric ratemaps for the two halves of a recording with a correlation >0.5 that were not already identified as either EBCs or HDCs.

Phasic vs. tonic firing: Previous reports identified striatal cells with phasic or tonic spiking properties[19]. Cell classification as phasic or tonic firing was based on post-spike suppression (PSS) and the proportion of interspike intervals (ISIs) shorter than 2 s (Prop$_{\text{ISIs}>2\,\text{s}}$). PSS quantifies the amount of time that it takes a cell to return to its mean firing rate following a spike. Using a 1 s window with 1 ms bins, the autocorrelation of each cell's spike train was computed and smoothed with a 25 ms Hamming window. The duration of time following a spike for the firing rate to reach the mean was taken as the PSS. The proportion of time that a cell spends in long interval ISIs provides a measure of the regularity with which a cell spikes and was measured as the summation of all ISIs >2 s divided by the total session length. Cells with Prop$_{\text{ISIs}>2\,\text{s}}$ above 0.4 were classified as phasically firing neurons, while cells with Prop$_{\text{ISIs}>2\,\text{s}}$ below 0.4 and PSS <100 ms were classified as high firing neurons. As noted in the main text, almost all EBCs and iEBCs (97%) fired phasically.

Mixed Gaussian models: Distributions of preferred orientation or preferred distance were modeled as mixtures of Gaussian distributions using varying orders from 1 to 10. The optimal model was identified as the model that minimized the Akaike information criterion. The local maxima in the probability distribution

function of the optimal model greater than the mean of the probability density function were identified and reported as the peaks of the distribution. This process was conducted on the data obtained from each individual animal (Supplementary Fig. 5f–i) and the distributions of peaks obtained from the individual animals were fit with Gaussian mixture models again allowing for a range of orders from 1 to 20 and identifying the optimal model as that which minimized the Akaike information criterion. The outcome of this approach is in line with fitting the distributions of preferred orientation and preferred distance for all EBCs with Gaussian mixture models and identifying the peaks of those models (compare Fig. 2d, h to Supplementary Fig. 5d, f).

**Statistics**. Non-parametric statistics were used for all post hoc comparisons as the normality of the distributions was not assumed. Two-sided Wilcoxon's signed rank tests were used with a $p$ value threshold of 0.01 for all comparisons. The values reported throughout the text are the 25th, 50th, and 75th percentile of the relevant distribution.

**Reporting summary**. Further information on research design is available in the Nature Research Reporting Summary linked to this article.

## Data availability
The data used in this study are available from the corresponding author upon reasonable request.

## Code availability
The custom written code used to analyze the data in this study is available from the corresponding author upon reasonable request.

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

## Acknowledgements
We would like to thank Andrew Alexander, Mark Brandon, and Jason Climer for helpful discussions and comments, as well as Britahny Baskin for technical assistance. This research was supported by National Institutes of Health NIMH R01 MH061492 and MH060013 and Office of Naval Research MURI grant N00014-16-1-2832 and N00014-18-S-F006.

## Author contributions
J.R.H. and M.E.H. designed the study. J.R.H. conducted all aspects of the experiments. J.R.H. and G.W.C. analyzed the data. J.R.H. and M.E.H. wrote the paper and G.W.C. provided feedback on the paper.

## Additional information

**Competing interests:** The authors declare no competing interests.

