## [Peer Review File · Nature Communications]

Editorial Note: This manuscript has been previously reviewed at another journal that is not operating a transparent peer review scheme. This document only contains reviewer comments and rebuttal letters for versions considered at Nature Communications .

Reviewers' Comments:

Reviewer #1:

Remarks to the Author:

Hinman et al. demonstrate that rat striatal neurons exhibit firing sensitive to the animal's orientation and distance from walls bounding an arena environment. The animals orientation and distance is given by its current trajectory as opposed to head position/orientation. Such "egocentric boundary cells" have not previously been reported and the cell type opens up an interesting set of questions that may well lead to an understanding of how sensing proximity to boundaries interfaces with motor control mechanisms to yield movement planning in accord with local affordances. The analyses are sound and the data plentiful.

As the manuscript is already the product of an extensive review/revision process, it is not surprising that it is essentially a finished product and appropriate for publication. There are, however, a couple of points to make that may aid in putting together the final product (see below).

1) In most of the examples given, it would be helpful to present the resultant found for the rate X orientation analyses. In figure 1, for example, it is difficult to make out that the four neurons with close-proximity egocentric border sensitivity have very different orientations.

2) In my opinion, the relationship to movement trajectory versus head direction is key. I would put the figure showing the resultant differences into the main manuscript. This data goes some way toward an argument that the tuning has more to do with context of movement than with visual sensing (in the absence of dark experiments). I wonder if the authors have tested the possibility that tuning (resultant lengths) might be stronger for the near future (100ms?) movement trajectory of the animal relative to boundaries.

Reviewer #2:

Remarks to the Author:

This is a revised manuscript detailing the discovery and testing of egocentric boundary cells recorded in the dorsomedial striatum. The authors generally responded well to all of my comments. However, there is one comment that the authors did not address very well and should be addressed before publication:

Point 8: I requested that the authors provide the number of sessions during which relevant cells were recorded. The authors provided this value for the rotation experiment, but not for any of the other manipulation experiments (visual change, size change, familiar environment, novel environment). The number of sessions conducted should be provided for all manipulations. This parameter gives a better sense of the consistency across experiments compared to the number of cells recorded because all cells could be responding the same way and the vast majority of sessions were presumably recorded with multiple cells.

Reviewer #3:

Remarks to the Author:

The authors have satisfactorily addressed all of my previous comments and I am happy to recommend this manuscript for publication. I have two very minor typographic suggestions (listed below) but I will leave these to the authors discretion. This is a great paper and it's a shame not to see it being published in [REDACTED], where it was originally submitted.

Abstract: "Here we present..." sound a little odd, maybe "Here we describe..." would be better?

Discussion: "...predictions from computational models [7,8] that propose that allocentric..." sounds a little odd, maybe "...predictions from computational models [7,8] which propose that allocentric..." would be better?

Reviewer #1 (Remarks to the Author):

Hinman et al. demonstrate that rat striatal neurons exhibit firing sensitive to the animal's orientation and distance from walls bounding an arena environment. The animals orientation and distance is given by its current trajectory as opposed to head position/orientation. Such "egocentric boundary cells" have not previously been reported and the cell type opens up an interesting set of questions that may well lead to an understanding of how sensing proximity to boundaries interfaces with motor control mechanisms to yield movement planning in accord with local affordances. The analyses are sound and the data plentiful.

As the manuscript is already the product of an extensive review/revision process, it is not surprising that it is essentially a finished product and appropriate for publication. There are, however, a couple of points to make that may aid in putting together the final product (see below).

1) In most of the examples given, it would be helpful to present the resultant found for the rate X orientation analyses. In figure 1, for example, it is difficult to make out that the four neurons with close-proximity egocentric border sensitivity have very different orientations.

We've added preferred angle (resultant angle) to each egocentric boundary ratemap in Figure 1 in order to facilitate an appreciation of the subtle tuning differences across cells.

2) In my opinion, the relationship to movement trajectory versus head direction is key. I would put the figure showing the resultant differences into the main manuscript. This data goes some way toward an argument that the tuning has more to do with context of movement than with visual sensing (in the absence of dark experiments). I wonder if the authors have tested the possibility that tuning (resultant lengths) might be stronger for the near future (100ms?) movement trajectory of the animal relative to boundaries.

The plot showing the resultant lengths when generating the egocentric boundary ratemaps using movement trajectory versus head direction is included in the main manuscript as Figure 1g. We considered the possibility that tuning might be stronger at non-zero time lags and a preliminary analysis hints that there is a retrospective component to EBC coding. However, the analysis ultimately raised numerous questions that would need to be addressed in order for our confidence in the result to reach a satisfactory level. This fact and our concern that including this finding would distract from the main points of the manuscript led us to not include the analysis in the current manuscript, although it is a question that we are interested in addressing in the future.

Reviewer #2 (Remarks to the Author):

This is a revised manuscript detailing the discovery and testing of egocentric boundary cells recorded in the dorsomedial striatum. The authors generally responded well to all of my

comments. However, there is one comment that the authors did not address very well and should be addressed before publication:

Point 8: I requested that the authors provide the number of sessions during which relevant cells were recorded. The authors provided this value for the rotation experiment, but not for any of the other manipulation experiments (visual change, size change, familiar environment, novel environment). The number of sessions conducted should be provided for all manipulations. This parameter gives a better sense of the consistency across experiments compared to the number of cells recorded because all cells could be responding the same way and the vast majority of sessions were presumably recorded with multiple cells.

We apologize for this oversight. When we were going through the minor points during the previous round of revisions we specifically went to the line mentioned in the comment and made a revision on that line, but we obviously failed to generalize the comment to the other relevant instances. We have now updated the manuscript with the number of sessions for all of the manipulations.

Reviewer #3 (Remarks to the Author):

The authors have satisfactorily addressed all of my previous comments and I am happy to recommend this manuscript for publication. I have two very minor typographic suggestions (listed below) but I will leave these to the authors discretion. This is a great paper and it's a shame not to see it being published in [REDACTED], where it was originally submitted.

Abstract: "Here we present..." sound a little odd, maybe "Here we describe..." would be better?

Discussion: "...predictions from computational models [7,8] that propose that allocentric..." sounds a little odd, maybe "...predictions from computational models [7,8] which propose that allocentric..." would be better?

The suggested typographic adjustments have been made.